# Identifying key genes in COPD risk via multiple population data integration and gene prioritization

**Afeefa Zainab** [1] *, **Hayato Anzawa** [1,2], **Kengo Kinoshita** [1,2] *

**1** Graduate School of Information Sciences, Tohoku University, Sendai, Japan, **2** Tohoku Medical Megabank Organization, Tohoku University, Sendai, Japan

* afeefa@sb.ecei.tohoku.ac.jp (AZ); kengo@tohoku.ac.jp (KK)

**Data Availability Statement:** All relevant data are within the manuscript and its Supporting Information files.

**Funding:** The Project was supported by JST SPRING Grant Number JPMJSP2114 AZ. Meta-

## Abstract

Chronic obstructive pulmonary disease (COPD) is a highly prevalent disease, making it a leading cause of death worldwide. Several genome-wide association studies (GWAS) have been conducted to identify loci associated with COPD. However, different ancestral genetic compositions for the same disease across various populations present challenges in studies involving multi-population data. In this study, we aimed to identify protein-coding genes associated with COPD by prioritizing genes for each population's GWAS data, and then combining these results instead of performing a common meta-GWAS due to significant sample differences in different population cohorts. Lung function measurements are often used as indicators for COPD risk prediction; therefore, we used lung function GWAS data from two populations, Japanese and European, and re-evaluated them using a multi-population gene prioritization approach. This study identified significant single nucleotide variants (SNPs) in both Japanese and European populations. The Japanese GWAS revealed nine significant SNPs and four lead SNPs in three genomic risk loci. In comparison, the European population showed five lead SNPs and 17 independent significant SNPs in 21 genomic risk loci. A comparative analysis of the results found 28 similar genes in the prioritized gene lists of both populations. We also performed a standard meta-analysis for comparison and identified 18 common genes in both populations. Our approach demonstrated that trans-ethnic linkage disequilibrium (LD) could detect some significant novel associations and genes that have yet to be reported or were missed in previous analyses. The study suggests that a gene prioritization approach for multi-population analysis using GWAS data may be a feasible method to identify new associations in data with genetic diversity across different populations. It also highlights the possibility of identifying generalized and population-specific treatment and diagnostic options.

## Introduction

Chronic obstructive pulmonary disease (COPD) is a lung disease that is a growing concern, as it is expected to worsen with the age of the population [1, 2]. COPD is highly prevalent,

analyses were performed using the supercomputer system at ToMMo, Tohoku Medical Megabank Organization, Tohoku University, supported by AMED (Grant Number: JP21tm0424601) KK. Additional support was provided by the Research Support Project for Life Science and Drug Discovery (BINDS) from AMED (Grant Number: JP22ama121019) KK.

**Competing interests:** The authors have declared that no competing interests exist.

making it the third leading cause of death worldwide [3]. Various environmental factors, including smoking characterize it, but as many as 25% of patients with COPD are non-smokers [4]. Several genetic associations linked with COPD disease risk have been identified in patients with COPD with no smoking history or environmental factors [5–7], suggesting that the identification of genetic factors can provide novel insights into disease pathogenesis and aid in diagnosis and treatment [8]. Lung function is crucial for predicting lung diseases such as COPD [9]. Consequently, fluctuations in lung function and the Forced Expiratory Volume per second ratio Forced Vital capacity (FEV1/FVC) are used as indicators to predict the risk of COPD. Several GWAS have identified significant genetic associations between COPD and lung function [10–14], most of which have been conducted in the European population; however, recently, data from other population GWAS are also becoming available [15, 16].

Although GWAS have identified significant risk loci, the central portion, which is almost 90% of the identified variants, constitute the non-coding regions, not causing any direct association with the protein-coding sequence [17]. However, the associated loci consist of multiple genes, which makes it harder to identify the causal and functional genes [17]. The statistical relationships between locus and trait and a functional understanding of the biology behind illness risk, however, have been proven to be challenging to reconcile for several reasons. Initially, when a locus is linked to a disease, it does not indicate which variant or variants at that locus are truly responsible for the link the referred to "causal variant" nor which gene or genes is impacted by the causative variant—the designated "target gene" [18]. Therefore, performing post GWAS interpretation of GWAS results is necessary to identify functional genes. This requires being extra cautious when performing GWAS in divergent populations, as some studies have shown a lack of interpopulation replicability, suggesting that some risk-associated alleles are population-specific [19–21]. Disease prediction can be enhanced, and essential insights into medical genetics can be gained by examining the relationship between genetic ancestry and phenotype in newly admixed populations [22]. This will result in identifying common functional genes among different populations and population-specific genes.

To overcome the existing challenges of identifying functional genes, we propose a multi-population gene prioritization approach that uses diverse population data integration as a post-GWAS analysis. This study will help identify functional interpretation between reported variants to identify candidate functional genes. Furthermore, post-GWAS gene prioritization helps identify functional and potentially causal variants. As suggested by Gallagher and Chen-Plotkin [18] it is more efficient to identify the functional interpretations of previously identified GWAS loci rather than search for new genetic loci to contribute to the knowledge of pathophysiology.

## Methods

In this study, we propose a post-GWAS gene prioritization approach to overcome the difficulties in interpreting GWAS results. In the usual approach, GWAS results from different populations combined by meta-analysis to increase statistical precision by increasing sample size. However, in our approach, we used a comparative post-GWAS method for analyzing GWAS results by individually performing post-GWAS analysis on each population study result using a specific reference genome of that population and comparing it later to identify unique and similar features across both population studies. It focuses more on the functional interpretation of GWAS results and obtaining meaningful information from GWAS data. We used SNP and gene prioritization to perform post-GWAS interpretation of each population GWAS results. An overview of the proposed approach as compared to the traditional meta-analysis method is presented in S1 Fig.

## Data description

The data used in this study were from Japanese and European populations. Summary statistical results for the Japanese population GWAS were obtained from the Tohoku University Medical Megabank Organization ToMMo (dataset ID: TGA000010). This study focused on identifying genetic loci associated with lung function in a Japanese population [16]. Analyses of the ToMMo CommCohort (n = 14,061) were used for the post-GWAS analysis. The summary statistics results for the European population were obtained from a study by Shrine *et al* [23] who analyzed the data for lung function from the SpiroMeta consortium for 79055 individuals. Both studies utilized linear mixed models implemented in BOLT-LMM [24] model for conducting GWAS, therefore data was comparable. The obtained association signals are directly comparable because the two studies use the same statistical model (linear mixed models) and software (BOLT-LMM), which provides a consistent framework for controlling for relatedness and population structure. Therefore, the two datasets were utilized for post-GWAS enrichment analysis in this study. Each sample was collected after obtaining informed consent, and detailed quality control has been described in previous studies [16, 23]. The data was reanalyzed in this study using the online software FUMA (Developed by: Kyoko Watanabe Update/ maintenance: Douglas Wightman (d.p.wightman[@]vu.nl) Dept. Complex Trait Genetics at VU University Amsterdam) (v1.3.8) [25] for gene prioritization.

## SNP and gene prioritization for a population by using FUMA

In this study, we propose an approach based on a specific LD structure and the reference genome of each population to perform multiple-population GWAS data analysis. The steps and analysis results are illustrated in Fig 3. To perform this analysis, GWAS summary statistics data from each study were used for post-GWAS analysis. First, genomic risk loci were redefined based on summary statistics results from the original survey of FUMA using reference genome data for each population from the 1000 Genomes Project [26]. The East Asian population (EAS) reference panel was used for Japanese GWAS data, and the European population reference panel (EUR) was used for European GWAS data analysis to compute the $r^2$ and minor allele frequency MAF. After defining genomic risk loci, independent significant SNPs were identified using $P < 5 \times 10^{-8}$ and $r^2 \geq 0.6$ (default setting) as the threshold from the summary statistics results of the GWAS. Furthermore, if pairwise SNPs from independent significant SNPs had $r^2 < 0.1$, they were defined as the Lead SNPs. The maximum distance between the LD blocks to be merged into the genomic risk locus was 250kb. To analyze each population individually, reference data for LD were taken from the 1000 Genomes phase3 European and East Asian populations for the European and Japanese datasets, respectively.

The lead SNPs, independently significant SNPs, and Genomic risk loci were defined for each dataset. ANNOVAR [27] was used to annotate SNPs. For gene prioritization, genetic mapping was performed using positional expression mapping, quantitative trait loci (eQTL) mapping, and chromatin interaction mapping collectively. In positional mapping, SNPs were mapped to genes based on the CADD scores, which were obtained using 63 functional annotations and were deleterious [28]. The threshold for a deleterious score is considered 12.37 [29]; therefore, positionally mapped SNPs were filtered for CADD > 12.37. We used CADD > 12.37 as it is the minimum threshold for pathogenic SNPs and has been used as a threshold for highly deleterious SNPs. Compared to variants with lower scores, it suggests that the variant is more likely to have a functional impact because it ranks in the top 1% of all scored variants. As in this study, our focus was on functional variants, and this threshold worked well. However, depending on the goal of the study, this threshold can be changed for example CADD > 10 to include the top 10% of predicted deleterious variants [30]. If the

distance between the gene and SNP was < 10kb within the genomic locus, then the genes were determined by the identified SNP. The second mapping method, eQTL mapping, was used, and if an SNP had any significant effect on gene expression, it was mapped onto that gene. We used GTEx v8 [31] GTEx v8 lung tissue for eQTL mapping, with a false discovery rate (FDR) < 0.05. Based on GTEx RNA-seq data, gene expression values are log2 transformed average reads per kilobase of transcript per million reads mapped (RPKM) per tissue type after being winsorized at 50. Third mapping, called chromatin interaction CI mapping, was performed using preprocessed Hi-C data for lung tissues from 14 tissue types [32]. Fit-Hi-C preprocessed significant loops were acquired from GSE87112 [32]. Transcription start sites (TSS) of genes that include regions that strongly interact with risk loci are overlapped by the promotor region window (250bp upstream and 500bp downstream from TSS, default value). For gene mapping, genes whose promoter regions overlap with interacting areas are utilized. Results from these three types of mapping were then utilized to get a prioritized gene list.

We included Major histocompatibility complex MHC region in our analysis which in most cases, can be excluded because of its highly polymorphic nature. We tried and checked how it affected the results, and we noticed that while excluding the MHC region, it can simply results but it loses some crucial immune signals that are associated with lung function for example *AGER* is one of the associated genes with lung function and it gets removed after excluding MHC region. As a result, the effects of removing the MHC have to be evaluated in light of the particular study questions and characteristics under investigation.

After analyzing both datasets individually, thus maintaining the uniqueness and originality of the study, prioritized genes and SNPs were compared to identify similar genes and SNPs in both datasets, as well as unique genes in each dataset. This comparative analysis identified common genes between population groups and novel signals due to differences in the ancestral LD structure. The criteria for the novelty of variants was defined as the list of SNPs identified through post-GWAS analysis using a multi-population gene prioritization approach, which has not been reported previously in the original study or the GWAS catalogue. This may include variants that could be present in the same LD block as the variants identified by previous studies but were not defined as lead SNPs in GWAS hits. The focus of this study is mainly to identify the protein-coding gene list that has a functional association with the disease or phenotype. The analysis generates a list of SNPs in LD with the original SNPs and defines new lead signals from that, which are then further used to create a list of significant SNPs. Therefore, it ensures that any SNP in LD with the original could be found to identify novel signals. After obtaining common and prioritized gene lists, pathway enrichment analysis was performed using ShinyGO 0.76.3 [33]. Significant KEGG pathways enriched by the identified genes were included in the study to determine the functional annotation of the identified SNPs, and genes with COPD risk were identified using this method.

## Meta-analysis

Meta-analysis to compare and verify the efficacy of the proposed approach was also performed. A random-effects meta-analysis was performed using METAL [34]. The Japanese and European datasets were preprocessed for the meta-analysis. To ensure homogeneity in allele orientation, this script preprocesses and harmonizes summary statistics from the European and Japanese GWAS datasets, preparing them for meta-analysis with METAL software. A detailed description of the data analysis and R script for pre-processing files is available in the S5 File and code availability section. The dataset was assigned columns to make the data uniform for meta-analysis. After obtaining the meta-analysis results, post-GWAS analysis was performed

using FUMA, and the same parameters were set for post-GWAS analysis except for only 1000 Genomes phase3 European population was used as a reference genome for mapping.

## Results and discussion

In this study, publicly available summary statistics data from two different population GWAS on lung function measurement FEV1/FVC were reanalyzed using the SNP annotation tool FUMA GWAS v1.3.8 Functional Mapping and Annotation of Genome-Wide Association Studies [25]. It is a platform used for annotating, prioritizing, and visualizing GWAS results. The data used in this study consisted of European GWAS results (n = 79055) [23] and Japanese GWAS results (n = 14061) [16]. Details of the method and data are described in the Methods section.

This study was designed to analyze GWAS data individually, maintaining the uniqueness of each study, after which a comparative analysis was performed to identify similarities and differences in the prioritized gene lists obtained from both population data analyses. The results obtained from the population data analysis using FUMA are summarized in Table 1. 20 independent and four lead SNPs were identified from the Japanese GWAS data. In the European GWAS data, 66 independent SNPs were identified, with 25 lead SNPs.

### Gene prioritization results for Japanese population GWAS analysis

The Japanese GWAS results were analyzed using FUMA, which identified three genomic risk loci, as described in detail in (S1 Table in S1 File). These risk loci identified 20 independent significant SNPs and four lead SNPs (S2 and S3 Tables in S1 File). These four lead SNPs consisted of one intergenic and three intronic SNPs. Out of the 20 independent significant SNPs, nine novel SNPs were identified (not reported in the GWAS catalog) (S3 Table in S1 File). A list of lead and independently significant SNPs was obtained. Three types of mapping were used to identify and prioritize the genes. Positional mapping identified 15 deleterious SNPs with CADD score $\geq$ 12.37 and 26 genes through expression quantitative trait loci eQTL mapping of lung tissues and whole blood from GTEx portal v8, no genes were mapped as a result of chromatin interaction mapping. There could be several possibilities because of which there was no chromatin interaction mapped SNPs found in the Japanese population data. One of these could be environmental factors. Evolutionary pressures and environmental influences can alter the chromatin landscape among populations, which may result in population-specific variations in gene regulation. It is also observed with the growing evidence and data that

**Table 1. Summary of post-GWAS analysis results.**

| Summary of the SNP prioritization results (FEV1/FVC) | | | |
|---|---|---|---|
| **Features** | **Japanese Population** | **European Population** | **Meta-Analysis** |
| | **N = 14061** | **N = 79055** | |
| Genomic risk loci | 3 | 21 | 21 |
| Lead SNPs | 4 | 25 | 24 |
| Ind. Sig. SNPs | 20 | 66 | 69 |
| Candidate SNPs | 319 | 2089 | 2346 |
| Positional mapped genes | 44 | 61 | 40 |
| Genomic risk loci | 3 | 21 | 41 |
| eQTL mapped genes | 33 | 35 | 69 |
| Total Mapped genes | 50 | 84 | 21 |

This table gives the summary of the post-GWAS analysis results for Japanese and European GWAS and meta-analysis.

environmental influences can alter epigenetic profiles via covalent chromatin modifications, for example, posttranslational histone modifications and DNA methylation, influencing cellular phenotype and gene expression [35]. It is also a probability that smaller effect sizes or variations in the underlying genetic architecture in the Japanese population prevented the statistical thresholds used, (FDR < 1e−6) suggested by Schmitt et al [32], to detect chromatin interactions from being satisfied. Use of lower thresholds may increase in number of chromatin interactions. A total of 50 genes were prioritized using these mapping techniques out of which nine genes were commonly present in both eQTL and positional mapping.

A summary of gene and SNP prioritization for the Japanese GWAS population is illustrated in Fig 1(A). 30 genes were reported in the original GWAS of the Japanese population for FEV1/FVC, of which eight genes were also confirmed among the 50 prioritized genes from this study, and we identified 42 novel gene candidates that were not reported previously in the original GWAS. In this study analysis, our focus was to identify the protein-coding genes only which explains the difference in several genes from the original study as normally GWAS variants lie in non-coding parts of genes while all 50 prioritized genes identified in this study are protein-coding genes out of which only eight overlaps with 30 genes reported genes from original GWAS. The inclusion of non-coding regulatory elements will further enhance the results and may have more overlapping genes with previous studies.

### Gene prioritization results for European population GWAS analysis

European post-GWAS analyses identified 21 genomic risk loci, 25 lead SNPs, and 66 independent significant SNPs. Detailed information is provided in the S1-S3 Tables in S2 File. Of the 25 lead SNPs, five SNPs were novel signals identified in this study that were not reported in the GWAS catalog previously. Out of 66 independent significant SNPs we identified 17 novel signals details of which are provided in S3 Table in S2 File. Genetic mapping revealed 84 prioritized genes of which 47 were deleterious SNPs with a CADD score ≥ 12.37 and 22 genes

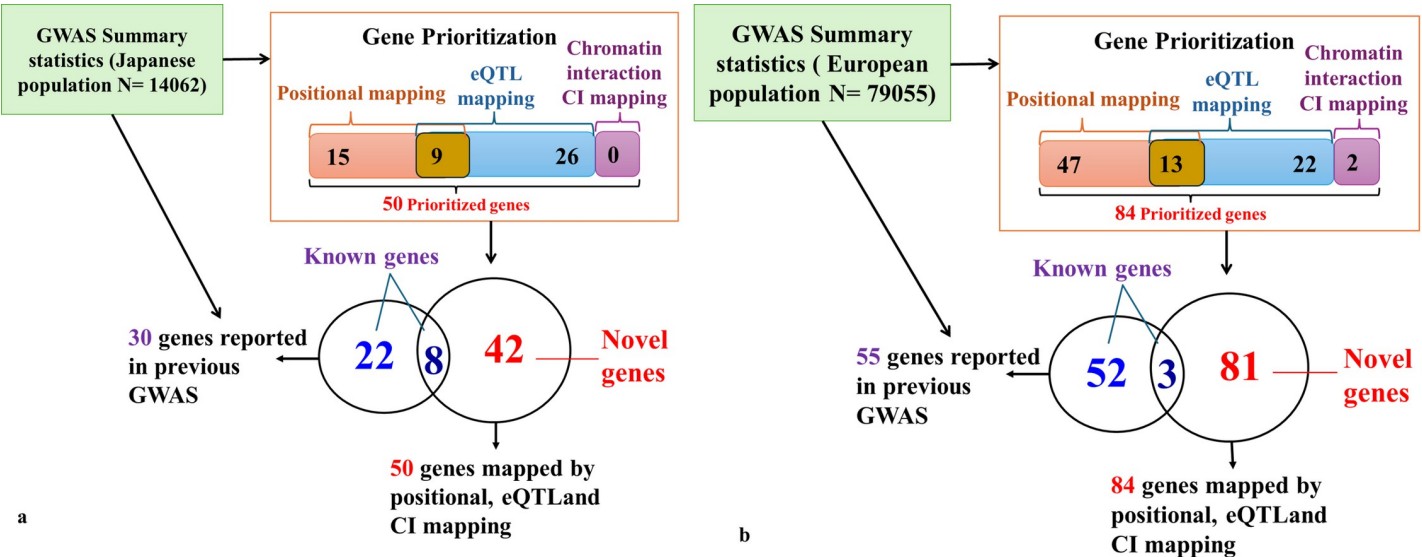

**Fig 1. Summary of SNP and gene prioritization results using gene mapping.** (a): Summary of SNP and Gene prioritization results using gene mapping in the Japanese population. Fifty genes were prioritized using positional mapping, eQTL mapping, and chromatin interactions in the Japanese population.42 novel genes were prioritized which were not reported in previous GWAS studies. (b) Summary of SNP and Gene prioritization results using gene mapping in the European population. Eighty-four genes were prioritized using positional mapping, eQTL mapping, and chromatin interaction mapping. Eighty novel genes were prioritized which were not reported in previous GWAS studies.

through expression quantitative trait loci eQTL mapping of lung tissues from GTEx portal v8, of these 13 SNPs were commonly mapped through both eQTL and positional mapping. A summary of the gene and SNP prioritization for the European population GWAS is illustrated in **Fig 1(B)**. A total of 55 genes were reported in the original GWAS of the European population for FEV1/FVC, of which three were present among 84 prioritized genes from this study, and we identified 81 novel gene candidates that were not reported previously in the original GWAS for FEV1/FVC. GWAS SNPs often fall in the non-coding region of genes but in this study, we focused on functional protein-coding genes and used them to explain the difference in the number of genes from the original study. 84 prioritized genes identified in this study are protein-coding genes, of which only 3 overlap with 55 genes reported genes from original GWAS. The number of overlapping genes with previous study could potentially be increased by the inclusion of non-coding regulatory elements. The observed variations between the European and Japanese populations regarding the number of lead SNPs, independent SNPs, candidate SNPs, and genomic risk loci in Table 1 can be due to several factors. Even though the difference in sample size is a key contributor because of its effect on statistical power, other factors such as variations in genomic architecture, allele frequencies, imputation accuracy, and gene-environment interactions all play important roles. Certain risk variations might be uncommon in Japanese populations but prevalent in European populations. Because common variants have a better statistical power to identify connections, GWAS are more likely to find associations with them. Alleles that were widespread in one group were sometimes not common in another, according to a recent thorough analysis of 3,873 genes from African, Asian, Latino/Hispanic, and European Americans [36]. Another possible factor can be population-specific alleles which can be due to variations in population history and evolutionary forces, some risk loci might be unique to or more common in European groups. There may be fewer identified risk loci in the Japanese data because these variants are either less common or non-existent in the Japanese population. This explain the differences in a number of alleles and loci found in Japanese and European population data.

In this study, a comparative post-GWAS analysis for both populations was performed to identify genetic similarities and differences between the populations. A summary of the study design and results is presented in **Fig 2**. Each population data was analyzed separately, and the results revealed a list of prioritized genes for each population. Out of 50 prioritized genes from Japanese population data and 84 genes from European data, there were 28 genes found to be common to both population groups, respectively.

## Significance of genes identified by multi-population gene prioritization approach

Comparative analysis of both population datasets identified novel signals, independent significant SNPs rs185396626 in the European dataset, and 6:32127198_G_A in the analysis of Japanese data that mapped with the dimethylarginine dimethylaminohydrolase 2 *DDAH2* novel gene (not reported in both population GWAS) and SNP rs185396626 was also identified by our approach and not reported previously in the GWAS catalog. The analysis showed that the populations exhibited genetic diversity, as different SNPs were identified for the same gene in different populations, providing evidence of population diversity. One such gene in this study the advanced glycosylation end-product-specific receptor *AGER* which is a protein-coding gene prioritized in both population datasets and mapped by rs2070600, rs2854050, and rs34422230 independent significant SNPs in European populations, and rs567657048; 6:32157364_G_A; and rs204993 in the Japanese population.

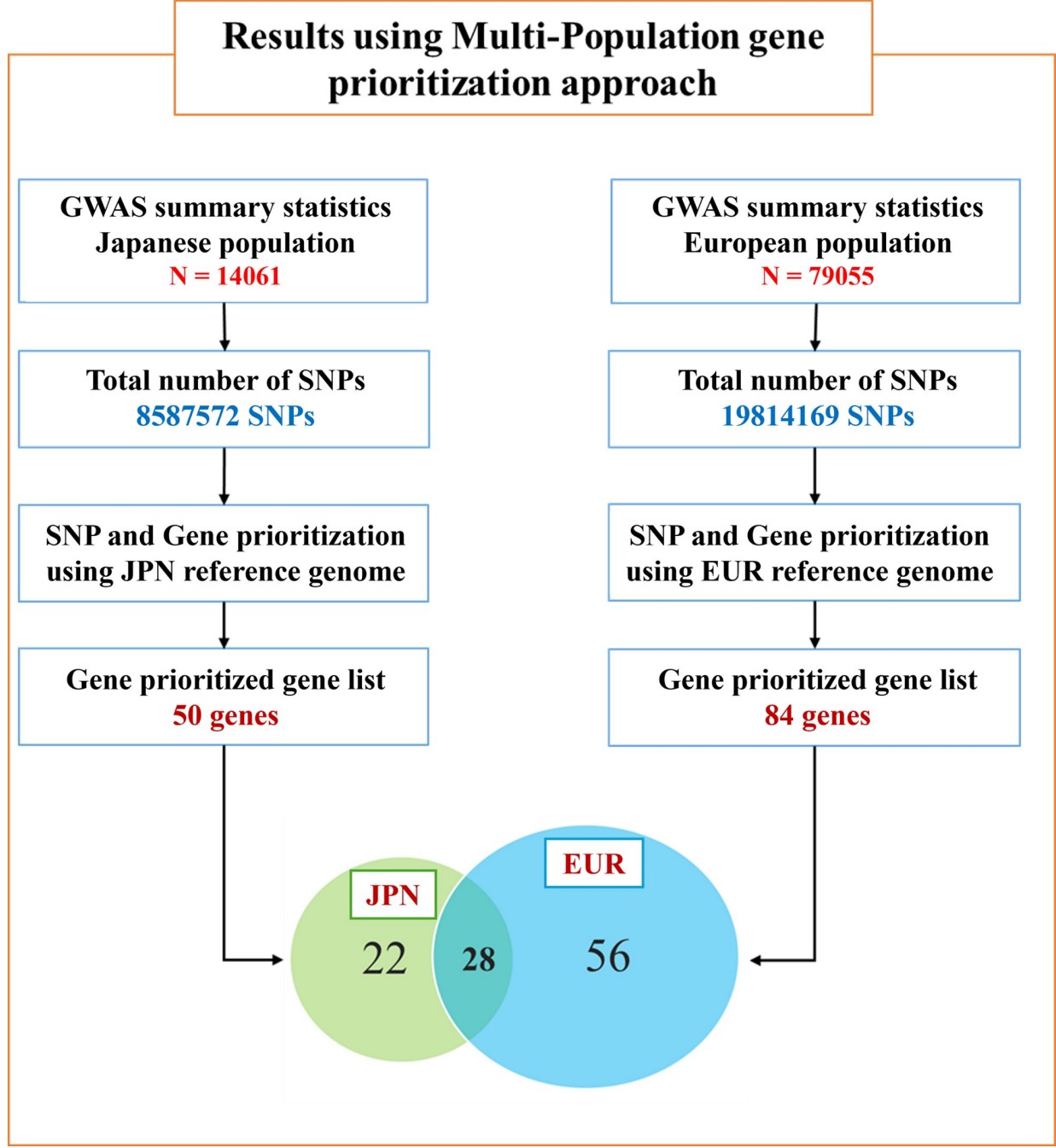

**Fig 2. Summary results of multi-population gene prioritization approach results.** Summary results of the multi-population gene prioritization approach results are illustrated in this diagram, with individual numbers of prioritized genes of each population and common genes among both population datasets.

To Identify the functional mechanisms and pathways associated with the prioritized genes, functional enrichment, and pathway analyses were performed for 28 genes common to both populations. In this study, we identified the *DDAH2* gene in the prioritized gene lists of both populations. This enzyme is also involved in the nitric oxide (NO) regulation pathway. It is involved in the hydrolysis of N(G), N(G)-dimethyl-L-arginine (ADMA), and N(G)-

monomethyl-L-arginine (MMA) which function as NOS inhibitors and thus regulate the production of NO [37]. There are some important immune regulatory pathways associated with respiratory diseases such as asthma and other important immune system regulatory pathways located in the Major histocompatibility complex (MHC) region enriched by these 28 genes and analyzed using ShinyGO (Shiny GO is developed and maintained by a small team at South Dakota State University (SDSU). 0.76.3 [33]. The plot illustrating the top KEGG pathways [38, 39] enriched by these genes is shown in **Fig 3**.The genes were found to be highly enriched in asthma, indicating a significant effects on pulmonary function and providing another evidence of risk association with the disease. Gene ontology GO enrichment results are given in the S1-S3 Tables in S4 File.

The identification of population-based genetic diversity is important for revealing the complex mechanisms involved in causing the disease. Several GWAS have been performed to identify the association between the risk of the disease and SNPs. One gene can often be regulated by multiple SNPs thus limiting GWAS-identified SNP as causal variants. GWAS reports the SNPs based on the most significant SNP identified; thus, it might not be able to detect significant signals that affect gene function [40]. Various challenges have been reported [41–43]. Therefore, it is important to perform post-GWAS to identify the linkages between SNPs and true causal genes and variants. There are several ways to perform post-GWAS. In this study, we performed cross-population post-GWAS to identify and prioritize significant genes and SNPs in each population, as well as to identify new independent significant SNPs and important phenotype-associated pathways and genes.

To further enhance the understanding of the protein functions and their interactions we checked and analyzed protein-protein interactions PPI using STRING database [44] for 28 common genes identified in this study. The results of PPI network showed that these proteins were closely linked to each other. We applied k-means clustering [45] to distinguish different

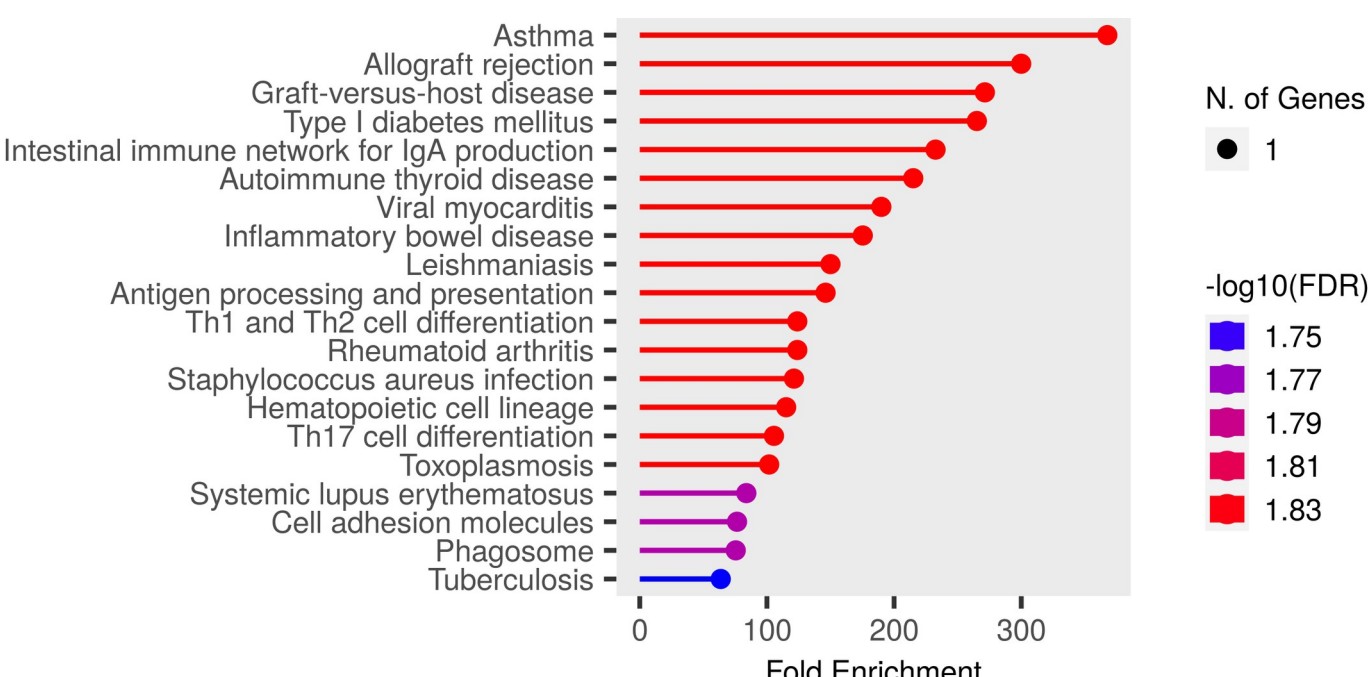

**Fig 3. Significant pathways enriched by prioritized genes.** The plot lists the top enriched pathways associated with the 28 common genes prioritized in both population datasets.

clusters. Three clusters were formed. Each cluster showed closed interaction and functional relationships among these genes. The PPI network is shown in Fig 4. Brief description and explanation of each cluster are as follows:

**Red cluster:** Proteins such as NOTCH4, PBX2, AGPAT1, AGER, GPSM3, and others are found in the red cluster. These proteins interact with one another both directly and indirectly. NOTCH4 seems to be a central protein in this cluster, with multiple connections to other proteins like PBX2, AGPAT1, and AGER. This suggests that NOTCH4 may play a central role in signaling pathways related to these proteins.

**Blue Cluster:** Contains proteins like HLA-DQA1, HLA-DQA2, HLA-DQB1, HLA-DQB2, and HLA-DRB5 that are associated with the human leukocyte antigen (HLA) system. The strong relationships between these proteins imply that their roles are closely connected, most likely involving immune response and antigen presentation. A highly interconnected function is suggested by the densely linked network of HLA genes. This is common for genes related to antigen presentation, where the proteins cooperate to form the immune response's major histocompatibility complex (MHC).

**Green Cluster**: Proteins including LY6G6D, LY6G6F, LY6G6C, LY6G5B, MPIG6B, PRRC2A, and others are found in the green cluster. Genes involved in cell surface signaling or immune control may be represented by this cluster. The tight clustering of the LY6 family genes (*LY6G6D*, *LY6G6F*, *LY6G6C*, etc.) raises the possibility that they play related or cooperative roles, maybe pertaining to the development or function of immune cells.

**TWIST2 and SKIV2L:** In the PPI network that is demonstrated in **Fig 4** these proteins seem to be separated and have few or no direct contacts. There may be little evidence of direct interactions with other proteins in this dataset, or their responsibilities may be unrelated to the primary functions of the other clusters.

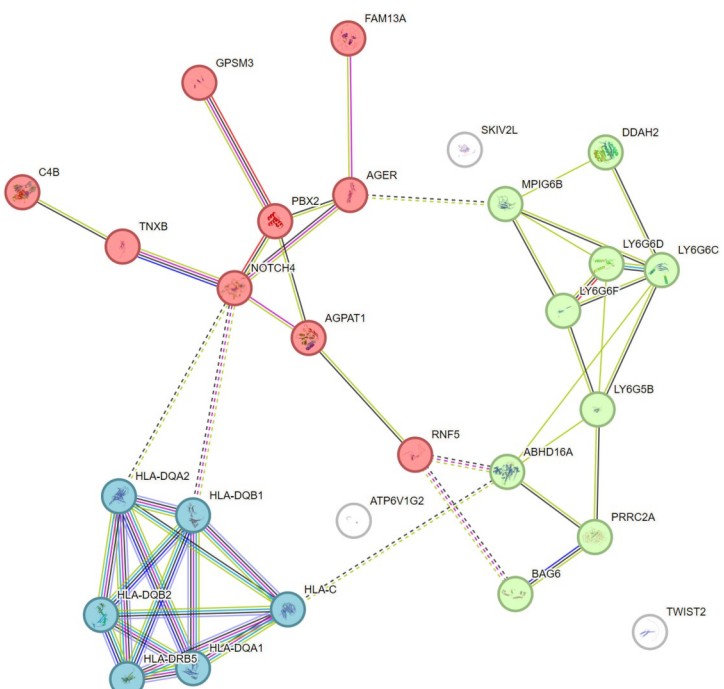

**Fig 4. PPI network of prioritized gene list of 28 genes created using STRING database.** The figure illustrates the PPI network of 28 identified genes and their interactions with each other. K-means clustering resulted in dividing proteins into 3 distinct clusters indicated by the colors red, blue, and green.

The PPI network analysis revealed that According to the clustering, many gene sets are involved in linked biological processes. To summarize the biological functions of each cluster, Adaptive immunity in particular is probably impacted by the blue cluster. Proteins implicated in transcriptional control and cell signaling may be represented by the red cluster. The green cluster may contain genes related to contacts or signaling on the cell surface, which is in line with the roles of proteins in the LY6 family. The k-means clustering has aided in classifying these proteins according to their patterns of interaction, exposing functionally linked gene sets that may be further investigated for their involvement in particular biological processes.

We identified new Lead SNPs in each population of the post-GWAS and identified novel signals and independent significant SNPs using positional and eQTL mapping. Pathway enrichment analysis identified important pathways, including the regulation of NO, which is an important factor in respiratory conditions, and the regulation of lung function. NO has both beneficial and deleterious effects associated with the respiratory system. Lung diseases, including COPD, asthma, and pulmonary fibrosis, show increased levels of NO [46]. In pulmonary fibrosis, increased NO levels are associated with elevated expression of *NOS2* in the alveolar epithelium [47]. Experimental studies have shown that the knockdown of *NOS2* results in a reduction in pulmonary fibrosis, indicating the role of NOS inhibition in the treatment of fibrosis [48].

In numerous physiological and pathological circumstances, nitric oxide (NO) plays a crucial role as a mediator of vasodilation [49]. Production and levels of NO are affected by several factors including the enzymes regulating NO production. Asymmetric dimethylarginine (ADMA) is one of the regulatory enzymes in production of NO, as it is the inhibitor of NOS which synthesizes NO. The dimethylarginine metabolic pathway is illustrated in **Fig 5**. People who have elevated levels of ADMA in their blood are more vulnerable to cardiovascular events and death [50, 51]. According to the pathway analysis, the two main enzymes that control natural ADMA concentration are PRMT and DDAH [37]. In both the systemic and pulmonary

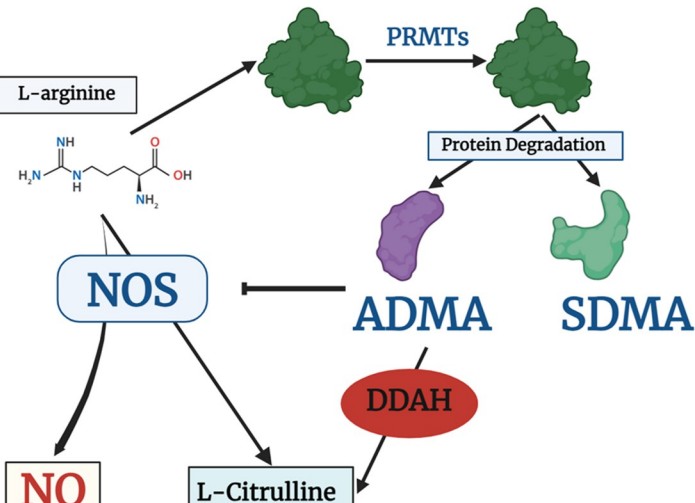

**Fig 5. Production and metabolic processes for dimethylarginine.** The diagram shows the production and metabolic processes for dimethylarginine. A family of protein arginine N-methyltransferases (PRMTs) is responsible for the (di-) methylation of protein-bound L-arginine residues, which results in the formation of dimethylarginines. As a result of physiological hydrolytic protein turnover, free ADMA and SDMA are released. While symmetric dimethylarginine (SDMA) does not directly affect NO synthase activity, asymmetric dimethylarginine (ADMA) decreases the production of nitric oxide (NO) from L-arginine. Dimethylarginine dimethylaminohydrolase (DDAH) can catabolically break down ADMA into L-citrulline and dimethylamine.

circulations, NO is the primary endothelial vasodilator mediator. Other endothelial mediators in arterial beds only play a substantial role when NO production is pharmacologically or genetically suppressed, or when NO signaling is compromised under pathological conditions [49]. There has been more research linking ADMA-induced deregulation of the NO system to pulmonary hypertension [52]. This links to the association with the risk of COPD.

This study also confirmed the results of previous GWAS, indicating that genes such as *AGER*, *TWIST2*, *NOTCH4*, and other similar genes were found and reported in both populations, thus providing evidence for their causality in association with FEV1/FVC. SNP identified as highly significant for *AGER* in the Japanese population GWAS were the exonic SNP rs2070600, whereas we identified novel signals (independent significant SNPs) for *AGER* rs567657048; 6:32157364_G_A; and rs204993, suggesting that the GWAS signal detected could be the result of the combined effect of these SNPs. However, in the European GWAS, AGER was not reported in the original study, but this study identified independent significant SNPs rs2070600, rs2854050, and rs34422230 through eQTL mapping of lung tissues for *AGER*. This also validates the finding that the Japanese GWAS SNP rs2070600 was mapped to the European population, depicting its significant association with the phenotype.

To ensure the efficacy of our approach, a random effect (RE) meta-analysis was performed using both population datasets and comparing the results with our approach. The meta-analysis results were then used to perform a post-GWAS analysis using the European population reference genome. The analysis identified 21 risk loci and 69 prioritized genes, of which 52 genes were common to the European population gene list, and 20 genes matched the Japanese population-prioritized gene list. However, 20 genes were common in the prioritized gene lists of both populations. Detailed information on identified risk loci of each population and meta-analysis is illustrated in S3 Fig.

The ancestral genetics and LD patterns of the reference genome play important roles in genetic associations in GWAS. This reduces the efficacy of trans-ethnic GWAS meta-analyses. In our approach, we mapped the population with their respective reference genomes, which provided new signals unique to each population group, as well as common signals showing genetic diversity, while using meta-analysis for the same data missed some significant population-specific associations. For example, in the post-GWAS analysis of the Japanese cohort, we found *LST1* which was not mapped in the European cohort and was neither mapped by meta-analysis.

The results of the meta-analysis showed that because of mapping with the European reference genome, genetic diversity was not fully identified. The Manhattan plots for each analysis result are shown in S2 Fig which also shows the similarity of meta-analysis results with European GWAS risk loci. We identified 28 unique genes commonly present in both population-prioritized gene lists using our approach, whereas the meta-analysis method only revealed 18 common genes. We identified 28 unique genes commonly present in both population-prioritized gene lists using our approach. The important genes identified using our approach included *XXbac-BPG32J3.20*, *LY6G6F*, *MEGT1*, *C6orf25*, *LY6G6C*, *DDAH2*, *ATP6V1G2*, *HLA-DQA1*, *HLA-DQB1*, *TNXB*, and *HLA-DRB5* which were not identified in the meta-analysis. Detailed comparative analysis and a list of prioritized genes by all cohorts using the Multi-Population gene prioritization approach and meta-analysis are mentioned in the S1-S5 Tables in S3 File. One of these genes is *DDAH2*, which was not present in the original GWAS and plays a significant role in the association with COPD. It is involved in the hydrolysis of asymmetric dimethyl-L-arginine (ADMA) and anti-monomethyl-L-arginine (MMA). Analysis of circulating ADMA in cardiovascular diseases revealed that ADMA levels were increased in pulmonary hypertension, hypoxia, heart failure, and conditions related to

hypercholesterolemia [53]. Telo *et al* [54] also observed elevated ADMA levels in patients with COPD pulmonary hypertension compared to a control group.

In this study, we mapped *TNXB* to independent significant SNP using positional mapping and the CADD score [29]. We discovered that a novel independent significant SNP, rs137893789, was mapped with TNXB with a CADD score of 15.21 in the European population GWAS using an LD-based specific population approach. This SNP has not been previously reported in a GWAS catalog. For the Japanese GWAS, 6:31933977_A_C, and 6:32015796_A_G were mapped with a CADD score of 15.77 using positional mapping with *TNXB*. This suggests that ancestral differences in the LD structure of each population result in missing heritability if not considered. Regional plots for *TNXB* in Japanese and European populations are shown in **Fig 6(A) and 6(B)**.

A comparison of our approach with a meta-analysis revealed that population diversity at the genetic level is a crucial factor when analyzing cross-population genetic data. A comparison of the results showed that the genetic loci and genes showed significant variation for each population group, depicting the unique LD structure for each population when mapped with the respective specific reference genome. Furthermore, our approach identified genetic similarities, such as the *AGER* gene mapped by rs2070600 in all cohorts, thus emphasizing the significant association of the gene with COPD in general for different ethnicities. Moreover, a meta-analysis, which is a promising approach for obtaining an overall combined effect for trans-ethnic GWAS, showed limitations in capturing the overall genetic diversity among both populations; for example, *DDAH2* was identified as a novel gene by our approach. Heterogeneity among different populations was recognized by identifying population-specific and common genes using our approach. Therefore, we suggest that to analyze trans-ethnic GWAS data, our approach can yield more promising results than meta-analysis.

## Conclusions

In this study, we utilized GWAS data, which is a commonly used method to identify risk-associated variants that play a crucial role in understanding complex genetic mechanisms [55]. Despite its potential, GWAS struggles with several limitations, including difficulties in integrating diverse population data due to ancestral linkage disequilibrium (LD) patterns and varying environmental, geographical, and genetic contexts and the majority of the identified variants present in the noncoding region thus making it hard to identify the functionally important risk associated genes.

The analysis of GWAS data extends beyond the identification of loci to understanding their association with disease mechanisms [56]. Post-GWAS functional analysis is critical for interpreting identified loci, as evolutionary forces create differences in allele frequencies and LD patterns among populations [57]. Therefore, distinguishing which linked variants are functional or causal is crucial [58, 59]. Even the variants that are present in the same LD block can have different functional associations with the risk of disease progression. Therefore, to understand complex underlying mechanisms of the diseases and biological processes post GWAS functional interpretation of identified risk variants is necessary.

Our study utilized GWAS results from Japanese and European populations regarding lung function measurement FEV1/FVC. We conducted post-GWAS analyses for each population, prioritizing genes and identifying independent significant SNPs and genes with functional association with the phenotype. However, our analysis limitation lies in the CADD score cutoff filter, which potentially misses tag SNPs. Future research should consider selecting all genes, not only protein-coding ones, and potentially discarding the CADD cutoff in SNP mapping. We wanted to focus on protein-coding genes mapped by deleterious SNPs in positional

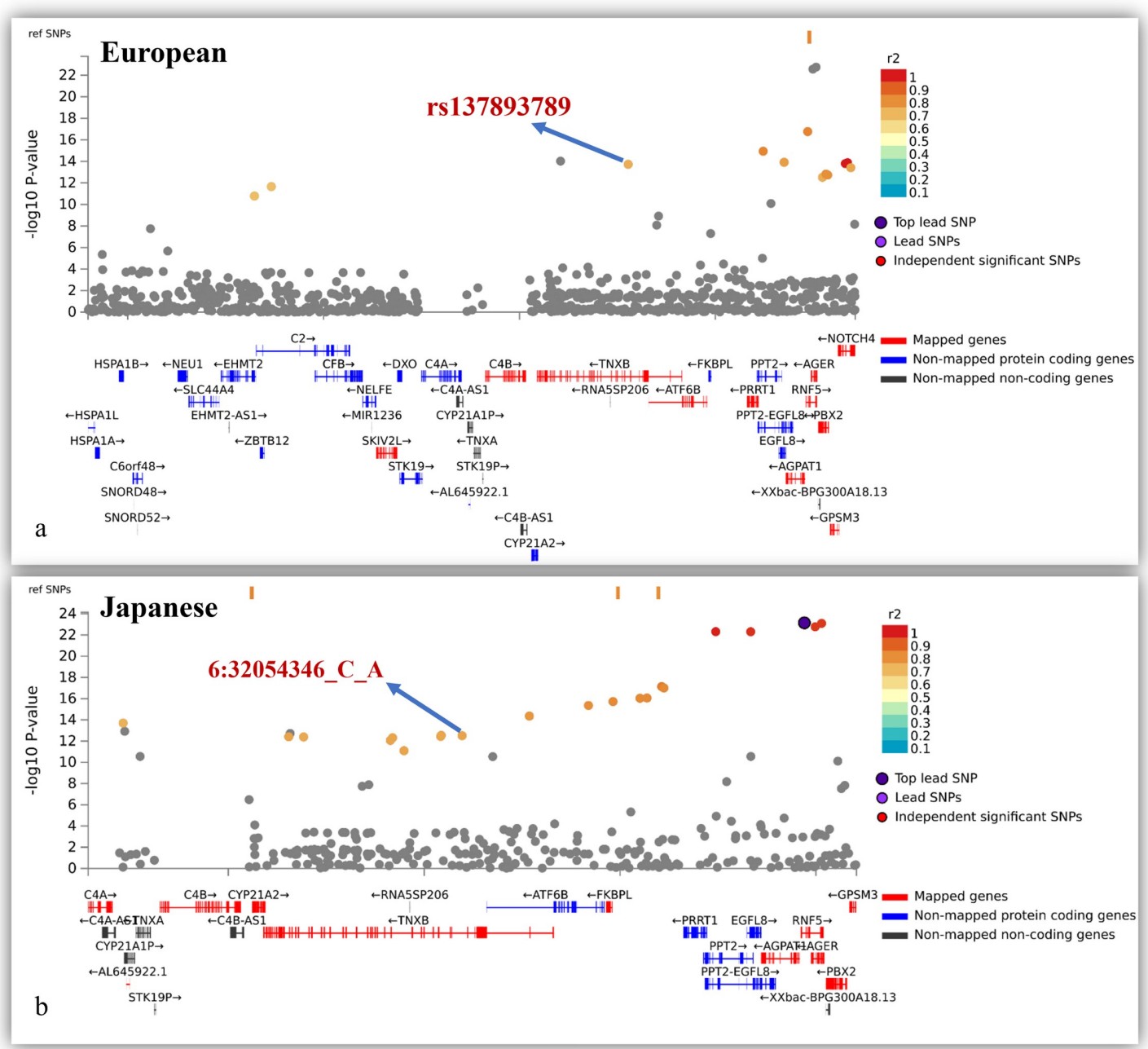

**Fig 6. Regional plot for chromosome 6 for Japanese and European population indicating risk associated independent significant SNPs mapped with TNXB.** (a) Regional plot for *TNXB* on chromosome 6 with independent significant SNPs in the European population. (b) Regional plot for *TNXB* on chromosome 6 with independent significant SNPs in the Japanese population.

mapping, therefore we chose the CADD filter. Despite not enhancing the sample size or statistical value of the study, our comparative post-GWAS interpretation and gene prioritization analysis offers an effective way to identify potential functional variants and genes related to disease pathways. It supplements GWAS results by incorporating expression data through eQTL and other gene prioritization details. It is possible to determine putative biological pathways behind disease relationships by integrating QTL maps with GWAS [17].

In conclusion, we identified 5 new lead SNPs, 26 new independent significant SNPs, and 122 unique genes in both populations through a trans-ethnic LD-based analysis and gene mapping. These findings, supported by a subsequent pathway analysis, offer new insights into the genetic association of lung function across populations. The results will further inform disease-associated variant mechanisms across multiple populations, potentially guiding the design of population-specific diagnostic and therapeutic studies.

## Supporting information

**S1 Fig. Overview of the meta-analysis and multi-population gene prioritization approach.** The meta-analysis approach (left) works by combining GWAS results from two populations (purple and green groups) into a single study through a merged linkage disequilibrium (LD) structure. The multi-population gene prioritization approach (right) keeps distinct GWAS and unique LD structures for each population for further analysis.
(TIF)

**S2 Fig. Manhattan plots showing some significant hits in Japanese, European, and meta-analysis GWAS results.** According to the meta-analysis (top-left) stronger signals are particularly seen at chromosomes 3, 4, and 17, While the Japanese GWAS (bottom) shows peaks on chromosomes 3 and 4, the European GWAS (top-right) highlights significant peaks on chromosomes 3, 5, and 17 (similar to meta-analysis). The genome-wide significance threshold is indicated by red dashed lines, which show how association signals vary among different populations.
(TIF)

**S3 Fig. The detailed information of the identified risk loci using European, Japanese, and meta-analysis results.** The y-axis shows the size of loci in kb, the number of SNPs in the loci, genes mapped within the risk loci, and a number of genes physically located in loci.
(TIF)

**S1 File. Results data consisted of identified significant loci S1 Table, lead SNPs S2 Table, and independent significant SNPs S3 Table in the Japanese population.**
(XLSX)

**S2 File. Results data consisted of identified significant loci S1 Table, lead SNPs S2 Table, and independent significant SNPs S3 Table in the European population.**
(XLSX)

**S3 File. Results data consisting of list of prioritized genes identified in Japanese population S1 Table, list of prioritized genes identified in European population S2 Table, list of prioritized genes identified through meta-analysis S3 Table, comparative analysis of multi-population gene prioritization approach results in S4 Table and comparative analysis of meta-analysis results with all cohorts in S5 Table.**
(XLSX)

**S4 File. This file contains the results of gene ontology GO enrichment analysis for 28 commonly identified genes.** S1 Table contains GO results for cellular components. S2 Table contains GO results for molecular function. S3 Table contains GO results for biological processes.
(XLSX)

**S5 File. This zipped file consists of code and data analysis files.** It consists of 2 zipped files,1_preprocessing.zip,2_Meta_analysis_parameters_METAL. The description of each file and its contents are mentioned in "Readme.txt".
(ZIP)

## Acknowledgments

We would like to acknowledge the Tohoku Medical Megabank Organization in Tohoku University, for letting us use a supercomputer system to perform our analysis for this study. We also appreciate and thank the reviewers for their valuable feedback and comments that helped us enhance quality of this manuscript.

## Author Contributions

**Conceptualization:** Afeefa Zainab.

**Data curation:** Afeefa Zainab, Hayato Anzawa.

**Formal analysis:** Afeefa Zainab.

**Funding acquisition:** Afeefa Zainab, Kengo Kinoshita.

**Investigation:** Afeefa Zainab, Kengo Kinoshita.

**Methodology:** Afeefa Zainab, Hayato Anzawa, Kengo Kinoshita.

**Project administration:** Afeefa Zainab, Hayato Anzawa, Kengo Kinoshita.

**Resources:** Afeefa Zainab, Kengo Kinoshita.

**Software:** Afeefa Zainab.

**Supervision:** Kengo Kinoshita.

**Validation:** Afeefa Zainab, Kengo Kinoshita.

**Visualization:** Afeefa Zainab, Kengo Kinoshita.

**Writing – original draft:** Afeefa Zainab, Hayato Anzawa, Kengo Kinoshita.

**Writing – review & editing:** Afeefa Zainab, Hayato Anzawa, Kengo Kinoshita.

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
