## [Decision Letter · Decision Letter 0]

15 Sep 2024

PONE-D-24-22709"Identifying Key Genes in COPD Risk via Multiple Population Data Integration and Gene Prioritization"PLOS ONE

Dear Dr. Zainab,

Thank you for submitting your manuscript to PLOS ONE. After careful consideration, we feel that it has merit but does not fully meet PLOS ONE’s publication criteria as it currently stands. Therefore, we invite you to submit a revised version of the manuscript that addresses the points raised during the review process.

We look forward to receiving your revised manuscript.

Kind regards,

Yonglan Zheng, Ph.D.

Academic Editor

PLOS ONE

Journal requirements: 1. When submitting your revision, we need you to address these additional requirements. Please ensure that your manuscript meets PLOS ONE's style requirements, including those for file naming. The PLOS ONE style templates can be found at https://journals.plos.org/plosone/s/file?id=wjVg/PLOSOne_formatting_sample_main_body.pdf and https://journals.plos.org/plosone/s/file?id=ba62/PLOSOne_formatting_sample_title_authors_affiliations.pdf. 2. We note that the grant information you provided in the ‘Funding Information’ and ‘Financial Disclosure’ sections do not match.  When you resubmit, please ensure that you provide the correct grant numbers for the awards you received for your study in the ‘Funding Information’ section. 3. Thank you for stating the following financial disclosure:  [The Project was supported by JST SPRING Grant Number JPMJSP2114AZ This project was also funded by Research Support Project for Life Science and Drug Discovery (Basis for Supporting Innovative Drug Discovery and Life Science Research (BINDS)) from AMED under Grant Number JP22ama121019.KK].  Please state what role the funders took in the study.  If the funders had no role, please state: ""The funders had no role in study design, data collection and analysis, decision to publish, or preparation of the manuscript."" If this statement is not correct you must amend it as needed. Please include this amended Role of Funder statement in your cover letter; we will change the online submission form on your behalf. 4. Thank you for stating the following in the Acknowledgments Section of your manuscript: [Meta-analyses were performed by using the supercomputer system in ToMMo, Tohoku Medical Megabank Organization in Tohoku University, which is supported by AMED under Grant Number JP21tm0424601. ]We note that you have provided funding information that is not currently declared in your Funding Statement. However, funding information should not appear in the Acknowledgments section or other areas of your manuscript. We will only publish funding information present in the Funding Statement section of the online submission form. Please remove any funding-related text from the manuscript and let us know how you would like to update your Funding Statement. Currently, your Funding Statement reads as follows:  [The Project was supported by JST SPRING Grant Number JPMJSP2114AZ This project was also funded by Research Support Project for Life Science and Drug Discovery (Basis for Supporting Innovative Drug Discovery and Life Science Research (BINDS)) from AMED under Grant Number JP22ama121019.KK]. Please include your amended statements within your cover letter; we will change the online submission form on your behalf. 5. Please amend either the abstract on the online submission form (via Edit Submission) or the abstract in the manuscript so that they are identical. 6. Please include captions for your Supporting Information files at the end of your manuscript, and update any in-text citations to match accordingly. Please see our Supporting Information guidelines for more information: http://journals.plos.org/plosone/s/supporting-information. 

Reviewers' comments:

Reviewer's Responses to Questions

**Comments to the Author**

1. Is the manuscript technically sound, and do the data support the conclusions?

Reviewer #1: Yes

Reviewer #2: Yes

2. Has the statistical analysis been performed appropriately and rigorously? 

Reviewer #1: Yes

Reviewer #2: Yes

3. Have the authors made all data underlying the findings in their manuscript fully available?

Reviewer #1: Yes

Reviewer #2: Yes

4. Is the manuscript presented in an intelligible fashion and written in standard English?

Reviewer #1: No

Reviewer #2: Yes

5. Review Comments to the Author

Reviewer #1: In this study, GWAS data from Japanese and European populations were utilized to investigate the association between lung function and genetic variants. Post-GWAS analyses were performed independently for each population cohort. Gene prioritization techniques were employed to identify genes of potential relevance, and independent SNPs that exhibited significant associations were identified. Furthermore, functional analyses were conducted to determine the functional associations between these identified genes and the lung function phenotype. The results revealed significant novel SNPs and genes that have not been reported by standard meta-analysis approach. These findings provided valuable insights into the multi-population GWAS, enhancing the understanding of the genetic landscape across divergent population.

Comment 1: In this study, three types of mapping approaches have been used to identify and prioritize the genes, with one of them as chromatin interaction mapping. In line 175, the authors stated that no genes were mapped as a result of chromatin interaction mapping in Japanese population, while two genes have been identified in European population. The authors should discuss the potential reasons for not identifying the genes in Japanese population. In Fig. 1, the results of chromatin interaction mapping in Japanese population should also be illustrated.

Comment 2: In Table 1, the Japanese population (N=14061) exhibits a significantly lower number of genomic risk loci, lead SNPs, independent SNPs, and candidate SNPs compared to the European population (N=79055). This discrepancy raises the question of whether the disparity is mainly due to differences in sample size or if other factors also contribute. The authors should discuss it comprehensively.

Comment 3: In line 234-236, the authors stated that they performed pathway analysis regarding 28 genes common to both populations. In Fig. 3, the results showed that the number of genes enriched for each pathway was around 2-3. The interactions among these genes were not illustrated thoroughly. Since the 28 genes are all protein-coding genes, protein-protein interaction network analysis should also be performed to better understand the functional relationships of genes in shared biological processes.

Comment 4: In line 244, the authors stated that the enrichment analysis was solely based on KEGG pathway. In general, the results of GO terms should also be provided. In addition, Supplementary Fig4 (line 245) was mistakenly cited as “the plot illustrating the top KEGG pathways associated with these genes”. The authors should correct it.

Comment 5: For Figure S1 and S2, the authors should provide detailed descriptions and explanations in the figure legends.

Comment 6:

Line 227: “The analysis showed that populations show” should be “The analysis showed that the populations exhibited”

Line 238: “Hydrolysis” should be “hydrolysis”

Line 245: “Genes were found” should be “The genes were found”

Line 246: “which indicates” should be “, indicating”;

“affect” should be “effects”;

“thus” should be “and”.

Line 252: “SNP” should be “SNPs”

Line 257-259: Should be changed into “There are several ways to perform post-GWAS. In this study, we performed cross-population post-GWAS to identify and prioritize significant genes and SNPs in each population, as well as to identify new independent significant SNPs and important phenotype-associated pathways and genes."

Line 263: “many” should be “both”

Line 280: "the regulatory enzyme" should be “the regulatory enzymes”, "which synthesize NO" should be "which synthesizes NO"

Line 287: “There's” should be “There has”

Line 334: “SNPs” should be “SNP”

Line 335-337: “We discovered…the European population GWAS.” should be “We discovered that a novel independent significant SNP, rs137893789, was mapped with TNXB with a CADD score of 15.21 in the European population GWAS using an LD-based specific population approach.”

Line 358-359: “Our study…mechanisms.” should be: “In this study, we utilized GWAS data, which is a commonly used method to identify risk-associated variants that play a crucial role in understanding complex genetic mechanisms.”

Line 368: “same” should be “the same”

Line 370: “underlaying” should be “underlying”

Line 385: “Lead” should be “lead”

Reviewer #2: This manuscript investigated the genetic variants associated with COPD risk, particularly focusing on the different ancestral genetic compositions across multiple datasets. The author first identified genes harboring variants associated with lung function decline within each GWAS cohort and then employed a “Multi-Population gene prioritization approach” to integrate results across studies. The authors reported 28 prioritized genes associated with the disease common across the populations and demonstrated the advantage of comparing the prioritized genes (in contrast to generating meta-analysis statistics) in identifying disease-associated biology when utilizing samples with diverse genetic backgrounds. The methods used in the paper were reasonable, and the paper was generally well written, and the method was described in detail.

Major comments

- The resolution of figures is low and it is hard to read the plot text. This is particularly true for Figure 1

- L56: “have been proven to be….”

- Please include a description of the model used for GWAS in each cohort, and whether they are directly comparable

- L120: is there a reason to choose a specific threshold of “CADD > 12.37”?

- L129: is only the lung data used?

- Several prioritized genes encode the MHC Class II molecules and are among the most polymorphic regions of the human genome. There have been GWAS studies that simply removed the MHC region. I believe this won’t affect the author’s results too much, but I wonder whether the author can comment on this, particularly how transethnic comparison may be affected by the MHC.

- L227-228: “The analysis showed that populations show genetic diversity, as different SNPs were 228 identified for the same gene in different populations” I think this is a very interesting observation. Is there previous work that makes similar findings? Does this attribute to the genetic background different, or potentially suggest different pathogenesis between populations?

- L277: This paragraph (and the later few as well) should belong to the discussion

- Code availability

Minor comments:

- While interchangeable, I believe “SNP” is more often used for GWAS, while SNV is usually used for somatic variants in the cancer studies

- Please make sure to use a consistent format for “trans-ethnic” or “transethnic”

6. PLOS authors have the option to publish the peer review history of their article (what does this mean?). If published, this will include your full peer review and any attached files.

Reviewer #1: No

Reviewer #2: No

---

## [Author Response · Author response to Decision Letter 0]

20 Oct 2024

Academic Editor

PLOS ONE

Dear Dr. Yonglan Zheng,

Thank you for reviewing our manuscript and providing valuable feedback. We have carefully considered your comments and those from the reviewers, and we have addressed them thoroughly.

We have included our responses to your comments as the Editor in the file named "Cover letter" and also sending it here. The detailed responses to the reviewers are provided in a separate file "Response to reviewers" for clarity. All the suggested changes and clarifications have been incorporated into the revised manuscript.

We look forward to your feedback and hope the revised manuscript meets your expectations.

Response: We have checked and considered the formatting guidelines and updated the manuscript's format according to the journal's requirements. 

Regarding the comments about the funding information from 2 and 4.

[Meta-analyses were performed by using the supercomputer system in ToMMo, Tohoku Medical Megabank Organization in Tohoku University, which is supported by AMED under Grant Number JP21tm0424601. ]

 [The Project was supported by JST SPRING Grant Number JPMJSP2114

AZ This project was also funded by Research Support Project for Life Science and Drug Discovery (Basis for Supporting Innovative Drug Discovery and Life Science Research (BINDS)) from AMED under Grant Number JP22ama121019.

KK].

Response: Thank you for clarifying and pointing out these points about the funding statement. 

The amended and updated funding statement is as follows :

[The Project was supported by JST SPRING Grant Number JPMJSP2114 AZ. Meta-analyses were performed using the supercomputer system at ToMMo, Tohoku Medical Megabank Organization, Tohoku University, supported by AMED (Grant Number: JP21tm0424601) KK. Additional support was provided by the Research Support Project for Life Science and Drug Discovery (BINDS) from AMED (Grant Number: JP22ama121019) KK.]

Kindly use this one in the funding information. 

 [The Project was supported by JST SPRING Grant Number JPMJSP2114

AZ This project was also funded by Research Support Project for Life Science and Drug Discovery (Basis for Supporting Innovative Drug Discovery and Life Science Research (BINDS)) from AMED under Grant Number JP22ama121019.

KK]. Please state what role the funders took in the study. If the funders had no role, please state: ""The funders had no role in study design, data collection and analysis, decision to publish, or preparation of the manuscript."" If this statement is not correct you must amend it as needed. Please include this amended Role of Funder statement in your cover letter; we will change the online submission form on your behalf.

Response: The funders did not have any role in the study design therefore following statement will be correct. 

Response: Thank you for pointing out this, we have updated the abstract so that the abstract on the online submission and the abstract in the manuscript are identical. 

Response: We have updated the manuscript by adding the captions of supporting information at the end according to journal requirements and format. 

Response: We have checked and updated the reference list so that all references are upto date and correctly cited.

best wishes 

Afeefa Zainab

For reference, we are also including the "Specific responses" for each reviewers comments as below:

Specific Comments by Reviewers

Reviewer 1

Thank you for taking the time to review our manuscript; we have incorporated the suggested changes and updated it. Kindly find detailed responses to individual comments as follows:

Comment 1: In this study, three types of mapping approaches have been used to identify and prioritize the genes, with one of them as chromatin interaction mapping. In line 175, the authors stated that no genes were mapped as a result of chromatin interaction mapping in the Japanese population, while two genes have been identified in the European population. The authors should discuss the potential reasons for not identifying the genes in the Japanese population. In Fig. 1, the results of chromatin interaction mapping in the Japanese population should also be illustrated.

Response: There can be several reasons for the lack of results in the Japanese population for chromatin interactions. Evolutionary changes and environmental factors can cause population-specific changes in the chromatin landscape. These factors can change epigenetic profiles, affecting gene expression and altering chromatin interaction signals. Furthermore, The Japanese population may have smaller effect sizes due to differences in different LD structures, which may not meet the strict statistical thresholds (e.g. FDR < 1e−6, as suggested by Schmitt et al. Cell Rep, 2016) required to identify significant interactions. More interactions might become visible if these thresholds are lowered.

 We have added descriptions in Lines 212-223 of the revised manuscript to clarify these points. To make the comparison with the European results more transparent, we have also updated Figure 1 to incorporate the chromatin interaction mapping results for the Japanese population. With these revisions, we think the reviewer's concerns are addressed, and a more thorough explanation is provided.

Comment 2: In Table 1, the Japanese population (N=14061) exhibits a significantly lower number of genomic risk loci, lead SNPs, independent SNPs, and candidate SNPs compared to the European population (N=79055). This discrepancy raises the question of whether the disparity is mainly due to differences in sample size or if other factors also contribute. The authors should discuss it comprehensively.

Response: Increased statistical power from a larger sample size facilitates the detection of relationships, particularly those with smaller effect sizes, which can be partly responsible for the discrepancy in the number of genomic risk loci, lead SNPs, and independent SNPs found. However, there can also be other factors that influence this variation. For example, certain variations may be easier to find in one population than another due to reasons such as variations in genetic diversity, linkage disequilibrium LD patterns, and allele frequencies among populations, which might affect the identification of loci and SNPs, in addition to sample size. Therefore, our approach also focused on the LD structures of each population. 

To clarify this point, we have added details and discussions to address your comment from lines 257-271.

Comment 3: In lines 234-236, the authors stated that they performed pathway analysis regarding 28 genes common to both populations. In Fig. 3, the results showed that the number of genes enriched for each pathway was around 2-3. The interactions among these genes were not illustrated thoroughly. Since the 28 genes are all protein-coding genes, protein-protein interaction network analysis should also be performed to better understand the functional relationships of genes in shared biological processes.

Response: Thank you for the excellent suggestion to include protein-protein interaction (PPI) network studies for the 28 frequent protein-coding genes. We conducted a PPI network analysis and included the results and description in the revised manuscript from lines 337-377, along with Fig 4 illustrating the PPI network generated using the STRING database. This added functional significance to the existing results as the relationship between the prioritized genes could be more apparent, which could help understand their mutual interactions.

Comment 4: In line 244, the authors stated that the enrichment analysis was solely based on KEGG pathway. In general, the results of GO terms should also be provided. In addition, Supplementary Fig4 (line 245) was mistakenly cited as “the plot illustrating the top KEGG pathways associated with these genes”. The authors should correct it.

Response: To clarify, we have added the findings of the Gene Ontology (GO) term enrichment analysis in the updated manuscript. The supplementary file (S4 file) and lines 318-319 contain a detailed explanation of these data, enabling a more thorough comprehension of the molecular activities, biological processes, and cellular components related to our results. It also gave significant insights about the genes in terms of regulating the immune system, such as positive regulation of the immune system, positive regulation of T cell activation, and regulation of cell-cell adhesion (S4 file, Table S3) can contribute to inflammation of lung diseases like asthma, chronic obstructive pulmonary disease (COPD), and other respiratory conditions. We think that this addition improves the analysis as a whole and successfully considers your feedback. We have also formatted the mistake in line 315.

Comment 5: For Figures S1 and S2, the authors should provide detailed descriptions and explanations in the figure legends.

Response: In supplementary information, we have added detailed descriptions for Figures S1 Fig and S2 Fig in lines 727-737. 

Comment 6 and other technical comments

Line 227: “The analysis showed that populations show” should be “The analysis showed that the populations exhibited”

Line 238: “Hydrolysis” should be “hydrolysis”

Line 245: “Genes were found” should be “The genes were found”

Line 246: “which indicates” should be “, indicating”; “affect” should be “effects”; “thus” should be “and”.

Line 252: “SNP” should be “SNPs”

Line 227: “The analysis showed that populations show” should be “The analysis showed 

Line 238: “Hydrolysis” should be “hydrolysis”

Line 245: “Genes were found” should be “The genes were found”

Line 246: “which indicates” should be “, indicating”;

“affect” should be “effects”;

“thus” should be “and”.

Line 257-259: Should be changed into “There are several ways to perform post-GWAS. In this study, we performed cross-population post-GWAS to identify and prioritize significant genes and SNPs in each population, as well as to identify new independent significant SNPs and important phenotype-associated pathways and genes."

Line 263: “many” should be “both”

Line 280: "the regulatory enzyme" should be “the regulatory enzymes”, "which synthesize NO" should be "which synthesizes NO"

Line 287: “There's” should be “There has” 

Line 334: “SNPs” should be “SNP”

Line 335-337: “We discovered…the European population GWAS.” should be “We discovered that a novel independent significant SNP, rs137893789, was mapped with TNXB with a CADD score of 15.21 in the European population GWAS using an LD-based specific population approach.”

Line 358-359: “Our study…mechanisms.” should be: “In this study, we utilized GWAS data, which is a commonly used method to identify risk-associated variants that play a crucial role in understanding complex genetic mechanisms.

Line 368: “same” should be “the same”

Line 370: “underlaying” should be “underlying”

Line 385: “Lead” should be “lead”

Response: All the above points are revised accordingly.

Reviewer 2:

Thank you for reviewing our manuscript and providing valuable feedback. We have provided a detailed explanation of each comment and incorporated the necessary changes.

Major comments

- The resolution of figures is low and it is hard to read the plot text. This is particularly true for Figure 1

Response: We have updated the figures with high resolution. 

- Please include a description of the model used for GWAS in each cohort, and whether they are directly comparable

Response: The models were comparable as both studies utilized linear mixed models implemented in BOLT-LMM. The European and Japanese population GWAS outcomes are directly comparable since the same statistical methodology was applied to both cohorts. Because any variation can be more positively attributed to variations in genetic architecture, LD structure, or allele frequencies rather than to differences in analysis methodology, this consistency enables a more trustworthy interpretation of the differences in detected genetic associations between the two populations.

In the revised manuscript, we have clarified the description of the models used for GWAS in each cohort, specifically in lines 97-101. We believe these additions make the model comparison more transparent and address your point.

- L120: is there a reason to choose a specific threshold of “CADD > 12.37”?

Response: The reason why we used CADD > 12.37 is that it is the minimum threshold for pathogenic SNPs and has been used as a threshold for highly deleterious SNPs. The variant is more likely to have a functional impact than variants with lower scores because it ranks in the top 1% of all scored variants. As in this study, our focus was on functional variants, and this threshold worked well. To clarify this point, we have added the following description in lines 130-136.

- L129: is only the lung data used?

Response: Yes, only the lung data was used. 

- Several prioritized genes encode the MHC Class II molecules and are among the most polymorphic regions of the human genome. There have been GWAS studies that simply removed the MHC region. I believe this won’t affect the author’s results too much, but I wonder whether the author can comment on this, particularly how transethnic comparison may be affected by the MHC.

Response: We agree that the MHC region can be excluded in most cases because of its highly polymorphic nature, but we tried to check how it affected the results. As a result, we noticed that excluding the MHC region can miss some crucial immune signals associated with lung function. For example, AGER is one of the genes related to lung function, and it is removed after excluding the MHC region. As a result, the effects of removing the MHC have to be evaluated in light of the particular study questions and characteristics under investigation. To clarify this point, we have added the following description in lines 149-155.

-L227-228: “The analysis showed that populations show genetic diversity, as different SNPs were 228 identified for the same gene in different populations” I think this is a ver

---

## [Editor Report · Decision Letter 1]

23 Oct 2024

Identifying key genes in COPD risk via multiple population data integration and gene prioritization

PONE-D-24-22709R1

Dear Dr. Zainab,

We’re pleased to inform you that your manuscript has been judged scientifically suitable for publication and will be formally accepted for publication once it meets all outstanding technical requirements.

Kind regards,

Yonglan Zheng, Ph.D.

Academic Editor

PLOS ONE

---

## [Editor Report · Acceptance letter]

28 Oct 2024

PONE-D-24-22709R1 

PLOS ONE

Dear Dr. Zainab, 

I'm pleased to inform you that your manuscript has been deemed suitable for publication in PLOS ONE. Congratulations! Your manuscript is now being handed over to our production team.

Kind regards, 

on behalf of

Dr. Yonglan Zheng 

Academic Editor

PLOS ONE